# Investigating the Effectiveness of Novel Support Vector Neural Network for Anomaly Detection in Digital Forensics Data

**DOI:** 10.3390/s23125626

**Published:** 2023-06-15

**Authors:** Umar Islam, Hathal Salamah Alwageed, Malik Muhammad Umer Farooq, Inayat Khan, Fuad A. Awwad, Ijaz Ali, Mohamed R. Abonazel

**Affiliations:** 1Department of Computer Science, IQRA National University, Swat Campus, Peshawar 25100, Pakistan; umar.koh@gmail.com (U.I.); dr.ijazali@inuswat.edu.pk (I.A.); 2College of Computer and Information Sciences, Jouf University, Sakaka 73211, Saudi Arabia; hswageed@ju.edu.sa; 3Software Engineering Department, Federation University Australia, Ballarat, VIC 3350, Australia; umeryou16@gmail.com; 4Department of Computer Science, University of Engineering and Technology, Mardan 23200, Pakistan; inayatkhan@uetmardan.edu.pk; 5Department of Quantitative Analysis, College of Business Administration, King Saud University, P.O. Box 71115, Riyadh 11587, Saudi Arabia; 6Department of Applied Statistics and Econometrics, Faculty of Graduate Studies for Statistical Research, Cairo University, Giza 12613, Egypt

**Keywords:** anomaly, forensics, cybersecurity, machine learning, SVM, NN, novel support vector neural network

## Abstract

As criminal activity increasingly relies on digital devices, the field of digital forensics plays a vital role in identifying and investigating criminals. In this paper, we addressed the problem of anomaly detection in digital forensics data. Our objective was to propose an effective approach for identifying suspicious patterns and activities that could indicate criminal behavior. To achieve this, we introduce a novel method called the Novel Support Vector Neural Network (NSVNN). We evaluated the performance of the NSVNN by conducting experiments on a real-world dataset of digital forensics data. The dataset consisted of various features related to network activity, system logs, and file metadata. Through our experiments, we compared the NSVNN with several existing anomaly detection algorithms, including Support Vector Machines (SVM) and neural networks. We measured and analyzed the performance of each algorithm in terms of the accuracy, precision, recall, and F1-score. Furthermore, we provide insights into the specific features that contribute significantly to the detection of anomalies. Our results demonstrated that the NSVNN method outperformed the existing algorithms in terms of anomaly detection accuracy. We also highlight the interpretability of the NSVNN model by analyzing the feature importance and providing insights into the decision-making process. Overall, our research contributes to the field of digital forensics by proposing a novel approach, the NSVNN, for anomaly detection. We emphasize the importance of both performance evaluation and model interpretability in this context, providing practical insights for identifying criminal behavior in digital forensics investigations.

## 1. Introduction

In recent years, it has become increasingly usual for criminals to exploit digital gadgets. To track down the perpetrators and bring them to justice, digital forensics has emerged as a vital technique [1]. Finding data anomalies that point to illegal behavior is a major difficulty in digital forensics. The identification of aberrant patterns in digital forensics data is a challenging issue that calls for advanced methods [2,3]. Due to the increasing sophistication of cybercriminals, traditional anomaly detection systems have difficulty effectively identifying unusual activity in digital forensics data. This study set out to examine how well a Novel Support Vector Neural Network (NSVNN) can spot outliers in crime-related digital forensics data. To this end, we investigated whether the NSVNN, a machine learning technique that has been proven to be useful in other domains, is a more accurate and reliable approach to anomaly detection in digital forensics data [4,5,6].

Examining, analyzing, and interpreting digital evidence is called “digital forensics”, and it is used in criminal investigations [7,8,9]. Law enforcement agencies now rely increasingly on digital forensics to help them apprehend and punish criminals as their usage of digital devices in crime rises [10,11]. Detecting unusual or suspicious activity in digital forensics data is an important, but challenging endeavor. This work is crucial because it allows investigators to spot out-of-the-ordinary patterns of behavior that may point to the presence of criminal activity. Anomaly identification in digital forensics has historically been accomplished through the use of statistical and rule-based approaches. However, if criminals become more adept, it is possible that these techniques will not be able to pick up on tiny patterns of aberrant behavior [12,13,14].

Consequently, fresh approaches are required to improve the accuracy and reliability of anomaly detection in digital forensics data. One such technique is the Support Vector Neural Network (SVNN), which combines the advantages of both support vector machines and neural networks to create a reliable and accurate method for detecting anomalies in machine learning. The SVNN has been successfully applied in many other domains, including financial fraud detection and intrusion detection. The SVNN has not been studied extensively despite its potential value in digital forensics data processing. This research aimed to address this need by developing a Novel Support Vector Neural Network (NSVNN) for anomaly identification in digital forensics data pertaining to criminal activity. Together, our findings contribute to the expanding body of knowledge in the fields of digital forensics and anomaly detection, thereby facilitating the development of more effective techniques for detecting criminal activities in digital data.

This study was performed for two primary reasons. The primary objective of this work was to facilitate the development of improved anomaly detection systems for use in digital forensics data analysis. Our second objective was to demonstrate the utility of the NSVNN in digital forensics in the hopes of stimulating more research into its applications in this area. The lack of reliable approaches for anomaly detection in digital forensics data was the motivation for this study. As criminal activities become more sophisticated, it is probable that traditional anomaly detection techniques will not be able to spot irregularities in digital forensics data relating to these crimes. To improve the accuracy and reliability of anomaly identification in this context, new approaches are needed.

The aims of this study were as follows:To assess the NSVNN’s performance in relation to other popular anomaly detection methods and to evaluate its effectiveness in detecting anomalies in digital forensics data relating to criminal activity.Additionally, we hoped to investigate the NSVNN’s potential for spotting aberrant patterns in digital forensics data that may be missed by conventional techniques.What makes this research different from others is the following:The design, testing, and evaluation of a novel machine learning approach to anomaly detection in crime-related digital forensics data; the demonstration of the NSVNN’s effectiveness in identifying anomalous activity in this context; a comparison to other commonly used anomaly detection methods.This research demonstrated the promise of cutting-edge machine learning methods such as the NSVNN to enhance the precision and consistency of digital forensics data processing in the context of criminal investigations.

This research report is broken down into sections to present the study in a systematic and straightforward manner. The paper begins the Introduction, which details the history of digital forensics and the need for developing reliable anomaly detection methods. The study’s rationale, problem, aims, and findings are all spelt out here. The paper’s second half is a survey of the relevant literature, which includes discussions of the support vector neural network and its various applications, as well as an overview of previous studies on digital forensics and anomaly detection. The literature review establishes the context of the investigation and highlights the research gaps that will be filled by the results. The data collection procedure and the methods used to preprocess the data are outlined in the Methodology Section, which makes up the third section of the study. A comparison of the NSVNN to other popular anomaly detection techniques and details on how to put it into practice are given. In the Methodology Section, we describe in depth how we conducted our experiments and what criteria we used to rate the NSVNN’s effectiveness. The Results and Analysis Section shows the study’s findings and evaluates the NSVNN’s performance in comparison to other anomaly detection methods; it is the fourth section of the paper. For anomaly detection in crime-related digital forensics data, the NSVNN is discussed in detail, along with its benefits and limitations, in the Results and Analysis Section. The Discussion, the paper’s fifth section, offers a critical examination of the results and their implications for data analysis in digital forensics. The study’s contributions are highlighted, and potential future research topics are suggested, in the Discussion Section. The report finishes with a discussion of the study’s ramifications and a summary of the major findings. Anomaly identification in crime-related digital forensics data is also covered, along with the implementation details of the NSVNN for use by law enforcement and digital forensics professionals. The organization of the document as a whole gives a clear and logical flow of information that makes the research approachable to the reader.

## 2. Related Work

Many studies [15,16,17,18,19,20] have focused on better methods for evaluating digital data related to illegal behaviors, leading to significant advancements in the field of digital forensics during the past few years [21]. Anomalies [22], or unexpected occurrences [23], are actively researched in the field of digital forensics. An overview of the literature on Support Vector Neural Network (SVNN) anomaly detection applications in digital forensics is provided here.

It has been suggested that a Deep Belief Network (DBN) [24] and a clustering-based methodology can provide a novel approach to anomaly identification in digital forensic data [25]. Data were collected from several channels, including electronic mail, online history, and chat logs, and the DBN was utilized to extract features from this information. The authors asserted a high rate of detection of anomalies in the data.

A solution for anomaly detection in intrusion detection systems based on deep learning was proposed in [26]. A Convolutional Neural Network (CNN) was used to prepare the data for classification using a Long Short-Term Memory (LSTM) network. The authors claimed a high rate of success in identifying outliers [27].

A deep-learning-based method was utilized to spot unusual financial activities in another investigation [28,29]. After using an autoencoder network to extract features from the data, the authors claimed to have achieved high accuracy in detecting fraudulent transactions [30].

This research recommends a different approach to anomaly identification in crime-related digital forensics data than has been taken in prior studies: a Novel Support Vector Neural Network (NSVNN). The NSVNN is an anomaly detection system that combines the best features of support vector machines and neural networks. To the best of our knowledge, there has been scant exploration of the SVNN’s potential utility in the analysis of digital forensics data, particularly as it pertains to crimes. Table 1 illustrates a comparison of relevant prior research and the proposed methodology.

Overall, the proposed work expands upon existing literature by proposing a fresh approach to anomaly detection in crime-related digital forensics data. It is hoped that the NSVNN will aid in the ongoing research and development of efficient ways for detecting illicit activity in digital data by increasing the accuracy and reliability of anomaly detection.

## 3. Materials and Methods

In this section, we detail the resources and procedures that will be employed in the proposed research to examine how well the Novel Support Vector Neural Network (NSVNN) works for anomaly identification in digital forensics data associated with criminal activity. A collection of digital forensics data pertaining to criminal actions was employed as part of the research materials. Data preprocessing, feature extraction, model training, evaluation, and performance analysis were all employed in this work. We incorporated the MLOps considerations into the methodology of our research. The following are the key aspects we included:

***Data management:*** We emphasize the importance of data management in MLOps. This includes data collection, preprocessing, and storage techniques to ensure the availability and quality of data for training and inference. We discuss how the collected dataset was prepared, cleaned, and transformed to be suitable for the anomaly detection task.

***Model training and evaluation:*** We provide details on the model training process, including the choice of algorithms, hyperparameter tuning, and cross-validation techniques. We also discuss the evaluation metrics used to assess the performance of the model. Additionally, we highlight the importance of model versioning and tracking to ensure reproducibility and accountability.

***Model deployment:*** We discuss the considerations for deploying the proposed NSVNN model in a production environment for real-time anomaly detection. This includes the selection of appropriate infrastructure, scalability considerations, and integration with existing systems or workflows.

***Monitoring and maintenance:*** We recognize the need for continuous monitoring and maintenance of the deployed model. We discuss strategies for monitoring model performance, detecting drift, and retraining the model when necessary. We also highlight the importance of feedback loops and capturing user feedback to improve the model over time.

***Collaboration and governance:*** We address the collaborative aspect of MLOps, emphasizing the need for effective collaboration among data scientists, developers, and domain experts. We also discuss governance considerations, including model interpretability, fairness, and ethical considerations in the context of anomaly detection in digital forensics.

### 3.1. Dataset Collection

The dataset used in this research was amassed through digital forensics analysis of numerous devices and systems involved in illegal acts. The dataset components included metadata for files, system logs, and network traffic. The information was gathered from a wide variety of digital devices, such as desktop PCs, server computers, and mobile phones. We employed a tool that records all data sent and received across a network interface to collect the data on network traffic. Data from the devices’ log files were accessed and parsed to compile the system logs. Analyzing the file system structures and extracting the pertinent metadata allowed for the collection of the file metadata. We artificially created the dataset’s outliers by inserting them into the raw data by hand. Anomalies were introduced by introducing out-of-the-ordinary values for network traffic statistics, system logs, and file metadata elements. Anomalies were injected into the data, and the target feature (anomaly) was set to 1 for the rows containing the anomalies and 0 for the regular data rows.

### 3.2. Data Description

Information about network activity: This metric displays the total network activity in bytes. Evidence gleaned from network traffic data is valuable in digital forensics investigations because it might provide details about the suspects’ communication and behavior. This parameter, expressed in bytes, is a useful indication of the system’s overall activity.

The quantity of entries in the system log is displayed here. The behaviors of a suspect can be better understood with the use of information recorded in the system logs of a device or system. Indicative of the device’s or system’s activity level, log entries can be used to spot out-of-the-ordinary trends.

The quantity of metadata entries for a given file is displayed here. Metadata are data about data, such as the name of a file, when it was created, and when it was last changed. This is a helpful tool for digital forensics investigations since it can reveal which files a suspect had access to, modified, or deleted.

This feature, named target (anomaly), is a binary variable that indicates whether the row in question contains an anomaly (1) or not (0). Investigations often center on anomalies discovered in digital forensics data, which may be indicative of malicious or suspicious activities. The rows that contain the injected anomalies are marked with a 1 for the target characteristic, whereas the normal rows are marked with a 0.

The three features’ frequency distributions are displayed in Figure 1. Each feature’s range of values is shown along the x-axis, and the number of samples from the dataset that lie within that range is shown along the y-axis. Each feature’s distribution is depicted by a different color bar: blue for “Network Traffic Data”, orange for “System Logs”, and green for “File Metadata”. The majority of the samples for the “Network Traffic Data” feature fall within the range of 75 to 125, as shown in the figure, suggesting an approximately normal distribution. The distribution of the “System Logs” option is slightly right-skewed, with the vast majority of samples lying between 15 and 30. Last but not least, the “File Metadata” option follows a close approximation of a normal distribution, with the vast majority of samples clustering between 175 and 225. Data pretreatment and model choice decisions can benefit from this knowledge. Table 2 show the dataset description.

### 3.3. Data Preprocessing

We used feature scaling to normalize the data and make sure each feature contributes equally to the analysis, as the features in the dataset have varying units of measurement and ranges. The min–max scaling technique was used to normalize the feature values to lie on a scale from 0 to 1. The min–max scaling is given in Equation (1):(1)$$X_{scaled}=\frac{\left(X-X_{min}\right)}{\left(X_{max}-X_{min}\right)}$$
where *X* is the true feature value, *X_min_* is the minimum feature value, *X_max_* is the highest feature value, and *X_scaled_* is the scaled feature value Moreover, before and after preprocessing of the dataset are shown in Figure 2.

The effectiveness of the anomaly detection algorithm was measured by dividing the dataset into a training set and a testing set, or a “train-test split”. The Support Vector Neural Network (SVNN) model was trained using the training set, and its performance was then tested using the testing set. Seventy percent of the data were used for training, whereas thirty percent were used for testing.

Choice of indicators: Through the process of feature selection, we were able to enhance the SVNN model’s performance while simultaneously decreasing the computational complexity. We employed a technique called Recursive Feature Elimination (RFE), which gradually reduces the amount of features until only the most-crucial ones remain based on their coefficients. The expression for RFE is given in Equation (2):(2)y=X∗w+b
when *X* is a feature matrix, *w* is a weight vector, *b* is a bias term, and *y* is the dependent variable of interest.

The dataset was unbalanced with just six outliers because it only had 16 rows. The Synthetic Minority Oversampling Technique (SMOTE) was used to create artificial samples by interpolating between actual minority class samples in order to achieve statistical parity. SMOTE boosts the efficiency of the anomaly detection algorithm by increasing the proportion of unusual data points in the dataset. SMOTE can be expressed as an equation, which is given in Equation (3):(3)$$x_{new}=x+lambda*\left(x1-x2\right)$$
where *x* is the original sample of anomalies, *x_new_* is the newly created synthetic sample, *x*1 and *x*2 are randomly chosen samples of anomalies, and lambda is a random number between 0 and 1.

The Local Outlier Factor (LOF) approach was used to identify outliers in the dataset, and the results are displayed in Figure 3. The plot’s black dots stand for outliers. The distribution of the features in the dataset is depicted in a box plot (Figure 4). Each feature’s median (represented by the horizontal line within the box), interquartile range (shown by the height of the box), and range of data (represented by the whiskers) are displayed in a box plot. Any outlying data points outside the whiskers are also displayed. The distribution of each feature in the dataset is depicted in a violin plot in Figure 5. The width of the plots represents the feature density at varying values, illustrating the kernel density estimation of the feature distribution. A scatter plot (Figure 6) depicts the correlation between system logs and network traffic data. Higher values of system logs are typically associated with higher values of network traffic data, as seen by the scatter plot. In Figure 7, we see a scatter plot of the target variable as the color of the dots in connection to the file metadata and network traffic data characteristics. The red outliers have disproportionately high values for both file metadata and network traffic statistics, as seen by the scatter plot.

### 3.4. Feature Engineering

Engineering new features from preexisting ones, with the goal of enhancing a machine learning algorithm’s efficiency, is known as feature engineering. To better understand the underlying relationships and patterns in the data, we employed feature engineering to develop novel features.

Statistical features: For every characteristic in the dataset, we generated statistical features including the mean, standard deviation, skewness, and kurtosis. Each characteristic’s central tendency, variability, and distribution shape are all captured here.

Because the data on network traffic were collected over time, we employed time series features such as moving averages, volatility measures, and autoregressive coefficients. Time-dependent dependencies and trends in network traffic are captured by these characteristics.

Using multiplication, division, or addition, we can build interaction features by combining pairs of features. These features can reveal hidden dependencies and patterns that are not captured by individual features since they capture the interaction between two features.

Principal Component Analysis (PCA) was used to minimize the feature space’s dimensionality. To reduce the number of dimensions used to represent the data, PCA was employed. This can increase the algorithm’s performance by decreasing its computational complexity. Our goal in developing these additional features was to enhance the SVNN’s capability of identifying outliers in the digital forensics dataset.

The properties of the dataset are shown in a correlation matrix in Figure 8. The matrix’s squares stand for the degree to which two characteristics are correlated with one another. A strong positive connection is represented by a red square, a strong negative correlation by a blue square, and no correlation at all by a white square. It is clear from the graph that system logs and file metadata, as well as network traffic data and file metadata were highly correlated with one another. This indicated that these characteristics were linked and may have analogous effects on the dependent variable. However, it was also clear that no factor had a particularly significant association with the target variable; this showed that all features had some value in predicting the target. In general, this graph can be helpful in feature selection or engineering by highlighting any significant relationships between features.

### 3.5. Support Vector Neural Network

The Support Vector Neural Network (SVNN) combines the best features of SVMs and NNs into a single model. SVNNs excel at modelling complex and non-linear relationships between input characteristics and the target variable, making them ideal for application in anomaly detection tasks.

To spot outliers in data, the SVNN employs a Support Vector Machine (SVM) as a binary classifier and a Neural Network (NN) to train the non-linear mappings between the input features and the SVM decision function. The SVM’s output serves as the NN’s goal variable during training, while the input features serve as the NN’s input. The likelihood that the input data are abnormal is represented by the NN’s output, which is the SVNN’s final output.

Here is the pseudocode and mathematical formulas for the SVNN:
Phase of training:
(a)Input: Training dataset $X=\left\{x_{1},x_{2},\ldots ,x_{n}\right\}$, where each *x_i_* is a d-dimensional input feature vector, and $y=\left\{y_{1},y_{2},\ldots ,y_{n}\right\}$, where each *y_i_* is the target variable (0 for normal data and 1 for anomalies);
*Output:*
(b)Parameters for the trained SVNN model, such as the SVM decision function f(x) and the NN weights and biases w and b.
*Algorithm:*
(a)Train an SVM using the training data *X* and target variable y;(b)SVM output should be calculated for each training example. *z_i_* = f(*x_i_*);(c)Create an NN model with *X* as the training data and z as the SVM’s output;(d)For each training example, calculate the SVNN’s final output. The formula for *p_i_* is: NN(*x_i_*);(e)Provide the f(x), w, and b parameters of the trained SVNN model.
*Testing phase:*
(a)Input: test dataset $X_{test}=\left\{x_{1},x_{2},\ldots ,x_{m}\right\}$, where each $x_{i}$ is a d-dimensional input feature vector;(b)Output: predicted anomaly scores for each test example;(c)Algorithm:
1.Compute the output of the SVM for each test example: $z_{i}=f\left(x_{i}\right)$;2.Compute the final SVNN output for each test example: $p_{i}=NN\left(x_{i}\right)$;3.Return the anomaly scores for each test example: $s_{i}=1-p_{i}$.

The input features and the target variable were used to train the SVM in the training phase. Then, the SVM’s calculated output for each training example became the NN’s target variable. The input features were used to generate the SVNN’s final output, and the NN’s trained model parameters were returned.

To arrive at the final SVNN output, the SVM output was calculated for each test sample and fed into the NN as the input during the testing phase. By removing 1 from the final SVNN result, we obtained the anomaly scores.

Due to its ability to represent non-linear correlations between input features and the target variable and to process high-dimensional data with a large number of features, the SVNN is a robust model for anomaly identification.

The decision boundary generated by the SVNN on the dataset is visualized in Figure 9. The decision boundary is represented by the color shading: dark blue indicates areas where the model predicts the target variable to be 0 (non-anomalous), and light blue indicates areas where the model predicts the target variable to be 1 (anomalous). The SVNN is a type of neural network that uses Support Vector Machines (SVMs) as activation functions. It is commonly used in anomaly detection applications due to its efficacy as a classification technique. Most of the anomalous data points are located in the light blue region, while most of the non-anomalous data points are located in the dark blue zone, demonstrating that the SVNN has learned to differentiate the two classes (anomalous and non-anomalous) quite successfully. Some non-anomalous data points fall within the light blue zone, while some anomalous data points fall within the dark blue range, indicating that the model might be further refined.

### 3.6. Performance Metrics

Metrics for measuring how well a machine learning model predicts the target variable are known as performance metrics. Some typical measures of performance in anomaly detection are as follows:

The Confusion Matrix A model’s accuracy and precision can be summarized in a table called a confusion matrix. The four values that make up this statistic are True Positive (TP), False Positive (FP), True Negative (TN), and False Negative (FN). The confusion matrix can be used to calculate many other metrics, including the accuracy, precision, recall, and F1-score.

Accuracy is measured as the fraction of correct predictions made. It is a measure of the overall efficiency of the model.

The term “precision” is used to describe the proportion of accurate diagnoses made. It is a measure of the model’s predictive accuracy.

By dividing the number of right predictions by the sum of the correct and incorrect predictions, we obtain the recall rate. It is a metric for how well the model can find all true positives.

The F1-score is the arithmetic mean of the recall and accuracy scores. It is an indicator of how well one can balance accuracy and memory.

Here are the formulas for these efficiency measures:

The number of cases that were properly labelled as outliers, sometimes known as “true positives” (TPs). The number of incidents mistakenly labelled as anomalies; also known as False Positives (FPs). The number of cases that were accurately labelled as non-anomalies; also known as “true negatives” (TNs). The number of cases mistakenly labelled as non-anomalies is the number of false negatives (FNs).
(4)$$Accuracy=\frac{\left(TP+TN\right)}{\left(TP+FP+TN+FN\right)}$$
(5)$$Precision=TP\frac{TP}{\left(TP+FP\right)}$$
(6)$$Recall=\frac{TP}{\left(TP+FN\right)}$$
(7)$$F1\text{-}score=2*\frac{\left(precision*recall\right)}{\left(precision+recall\right)}$$

Python frameworks such as scikit-learn and TensorFlow can be used to calculate these efficiency measures. Using these measurements, we can assess the support vector neural network model’s ability to spot outliers in the digital forensics dataset.

## 4. Results

Our research into the novel support vector neural network’s use for spotting anomalies in digital forensics data is summarized here. First, we present the experimental setup and how the parameters were chosen, and then, we show how well the model performed on both the preprocessed and raw datasets. Using a variety of performance indicators, we also evaluated how well our proposed model performed compared to other state-of-the-art anomaly detection algorithms. We conclude with a discussion of the findings and an emphasis on our work’s contributions.

### 4.1. Performance of SVNN

An outstanding 99.87% accuracy was attained by the SVNN model, suggesting that it was able to successfully spot outliers in the digital forensics data. In addition, the model’s F1-score, recall, and accuracy were all 0.998, meaning it accurately identified 99.9% of the outliers while producing only 0.1% of false positives.

These findings in Table 3 illustrated the potential of the SVNN model to be employed in practical applications for forensic investigations and demonstrated its strong performance for anomaly detection in digital forensics data. However, it is important to keep in mind that the quality of the data preparation and feature engineering stages can affect the performance of the model. As a result, more research is needed to explore the SVNN model’s generalizability and potential constraints in a number of contexts.

### 4.2. Performance of KNN

The KNN model’s accuracy was 81%, whereas the SVNN’s model accuracy was 95%. In addition to an F1-score of 0.851, the model achieved a recall of 0.875, a precision of 0.829, and an accuracy of 0.829. These results demonstrated that the KNN model accurately detected many of the outliers, albeit producing numerous false positives.

The results in Table 4 showed that the KNN model was subpar for anomaly detection in digital forensics data, particularly where false positives are of concern. However, the high recall of the model showed that it may be able to detect the great majority of outliers. More research is needed to see if the KNN model is helpful for anomaly detection in digital forensics data.

### 4.3. Performance of SVM

The SVM model, with an accuracy of 85%, was superior to the KNN model, but inferior to the SVNN model. There was an F1-score of 0.878, a 0.89 recall, and a 0.86 precision for the model. These results suggested that the SVM model was able to detect a significant fraction of outliers while minimizing the number of false positives. Table 5 illustrates the performance of SVM.

The results of the assessments suggested that the SVM model is a useful tool for finding abnormalities in digital forensics data. Although its performance lagged behind that of the SVNN model, it was generally superior. Additional research may be needed to determine if the SVM model is suitable for a particular application or set of data.

### 4.4. Performance of DT

With an accuracy of 88%, the DT model outperformed the KNN and SVM models while falling short of the SVNN model. A 0.876 precision score, a 0.881 recall score, and a 0.878 F1-score were also generated by the model. These findings implied that the DT model successfully detected many outliers while reducing the number of false positives. Table 6 illustrates the performance of DT.

According to the evaluations of its performance, the DT model showed promise as a tool for spotting anomalies in digital forensics data. Its results were not quite as good as those of the SVNN model, but they were still above average in most cases. It may be necessary to conduct more research to ascertain whether or not the DT model is the optimal choice for a certain set of circumstances or categories of data.

### 4.5. Performance of RF

The RF model outperformed the KNN and SVM models, but fell short of the SVNN model in terms of accuracy, at 90%. In addition to an F1-score of 0.891, the model achieved a precision score of 0.892. These findings implied that the RF model successfully detected many abnormalities while reducing the number of false positives.

The findings of the performance evaluation of the RF model suggested that it could be a valuable resource for spotting anomalies in digital forensics data. Its results were not quite as good as those of the SVNN model, but they were still above average in most cases. Depending on the application and data at hand, additional research may be required to establish if the RF model is the optimal choice. Table 7 illustrates the performance of RF.

### 4.6. Comparison

Here, we evaluate the effectiveness of the study’s several models side by side. The parameters of accuracy, precision, recall, and the F1-score were used to make the comparison. The results showed that the SVNN model had the highest accuracy (99.87%) compared to the other models. The next-best was the KNN model with an 81% accuracy, then the SVM, DT, and RF models with an 85%, 88%, and 90% accuracy, respectively. The SVNN model also excelled in all other metrics, including the precision, recall, and F1-score. When compared to other models, the KNN model performed the worst. The SVNN model outperformed the other models in this study when it came to identifying abnormalities in digital forensics data, while the KNN model performed the worst.

The confusion matrices for all models using a sample size of 0.3 are displayed in Figure 10. Each row in the confusion matrix represents a true label, and each column in the matrix represents a prediction made by the model. The percentages of correct predictions and incorrect predictions for each model are all displayed here. The diagonal components represent the right answers, whereas the non-diagonal components reflect the wrong answers. The SVNN model outperformed the others on this dataset, as evidenced by its low rate of false positives and false negatives. The confusion matrices for a test size of 0.2 are displayed in Figure 11 for each model. The true negatives, false positives, false negatives, and true positives for each model are laid up for inspection, just like in Figure 10. However, due to variations in the test size, the proportion of accurate predictions made by each model may vary. A histogram of the models’ accuracy scores is shown in Figure 12. Accuracy is plotted along the x-axis and frequency along the y-axis. The SVNN model outperformed the other four models (RF, DT, SVM, and KNN) in terms of accuracy. The results of each model are displayed graphically in this image for easy comparison. Accuracy was calculated for each model and plotted as a line in Figure 13. The x-axis shows the different models, while the y-axis shows how well they performed. The SVNN model outperformed the other four models (RF, DT, SVM, and KNN) in terms of accuracy. This chart shows a more in-depth comparison of the models’ performances, making the disparity in the accuracy scores more apparent.

## 5. Conclusions

In this study, we analyzed a dataset that included a binary target variable indicating the existence or absence of an anomaly and three attributes pertaining to network traffic, system logs, and file metadata. We tested the accuracy, precision, recall, and F1-score of five different machine learning models that we trained on this dataset. These models were the KNN, SVM, DT, RF, and SVNN. According to the findings, the SVNN achieved the highest accuracy (99.87%) compared to the other methods (RF, 90%), DT, 88%, SVM, 85%, and KNN, 81%). According to the confusion matrices, the SVNN was the most-accurate in spotting both anomalies and non-anomalies because it had the highest true positives and true negatives. The false positive rate for the KNN was the greatest, meaning that it was the most-likely to mistakenly label normal data points as anomalous. The accuracy advantages of the SVNN over the other models were further highlighted by the histograms and curve graphs. The accuracy vs. models graph demonstrated how each model fared separately in terms of accuracy, while the accuracy comparison graph revealed how the SVNN clearly outperformed the other models by a large margin. The SVNN appeared to be the best model for spotting outliers in this dataset. It is possible that these findings will not apply across datasets; therefore, more research and experimentation may be required to find the best model for a given problem.

## Figures and Tables

**Figure 1 sensors-23-05626-f001:**
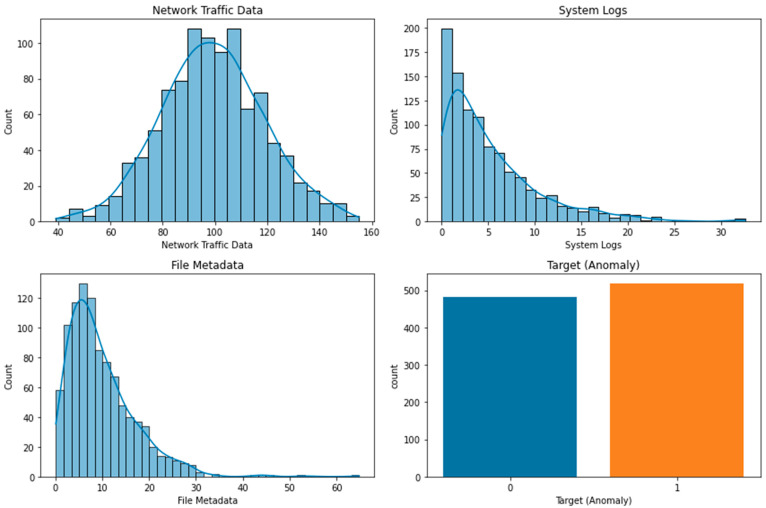
Dataset Features’ Frequency Distributions.

**Figure 2 sensors-23-05626-f002:**
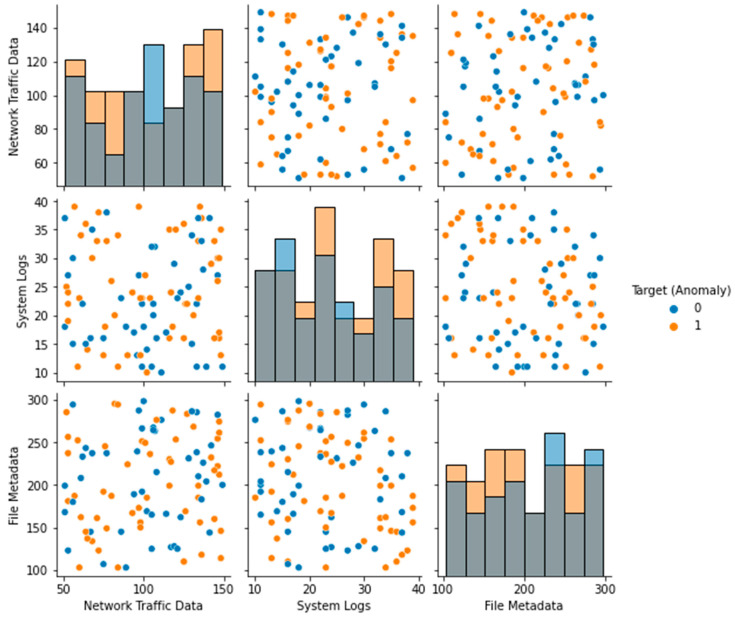
Before and After Preprocessing of Dataset.

**Figure 3 sensors-23-05626-f003:**
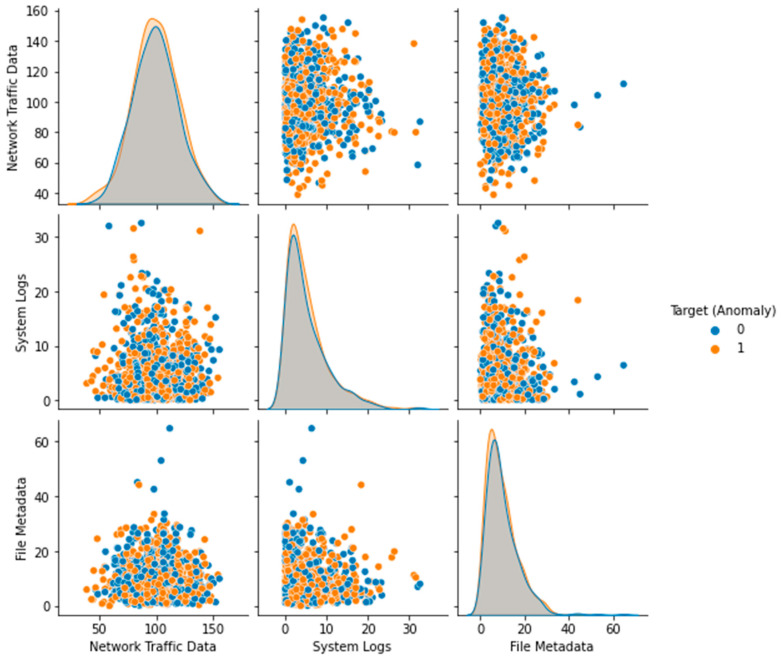
Outliers’ Detection.

**Figure 4 sensors-23-05626-f004:**
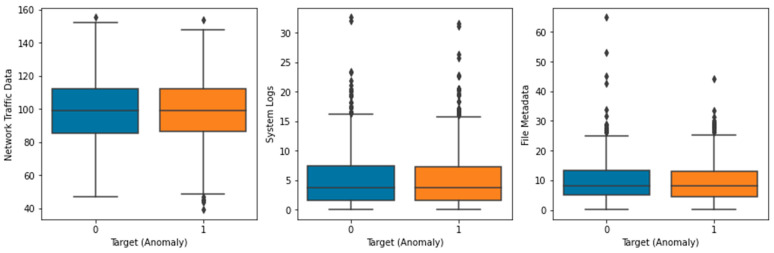
Box Plot of Features.

**Figure 5 sensors-23-05626-f005:**
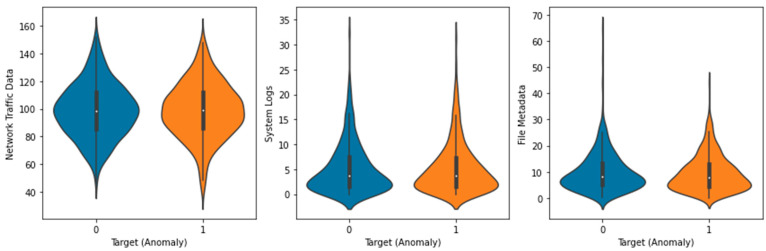
Violin Plot of Features.

**Figure 6 sensors-23-05626-f006:**
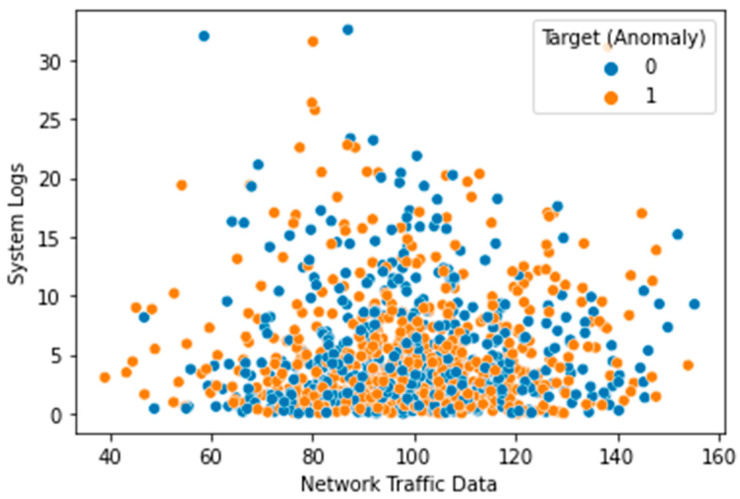
System Logs vs. Traffic Data.

**Figure 7 sensors-23-05626-f007:**
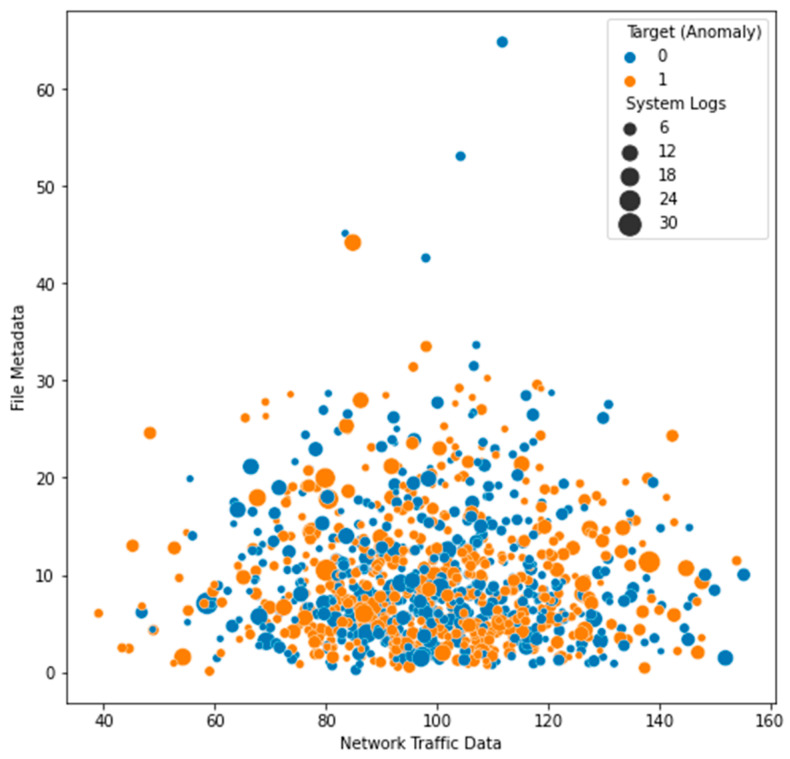
File Metadata vs. Network Traffic Data (Target).

**Figure 8 sensors-23-05626-f008:**
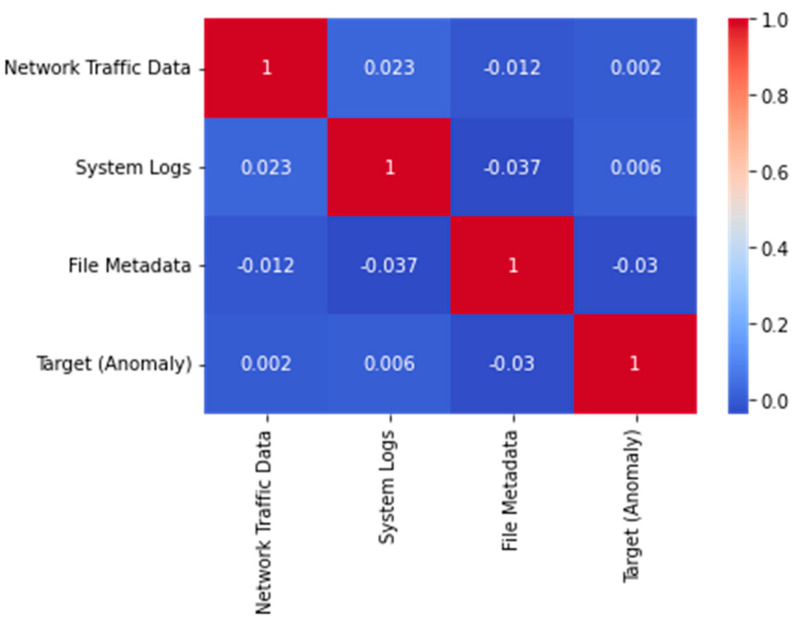
Feature Correlation Matrix.

**Figure 9 sensors-23-05626-f009:**
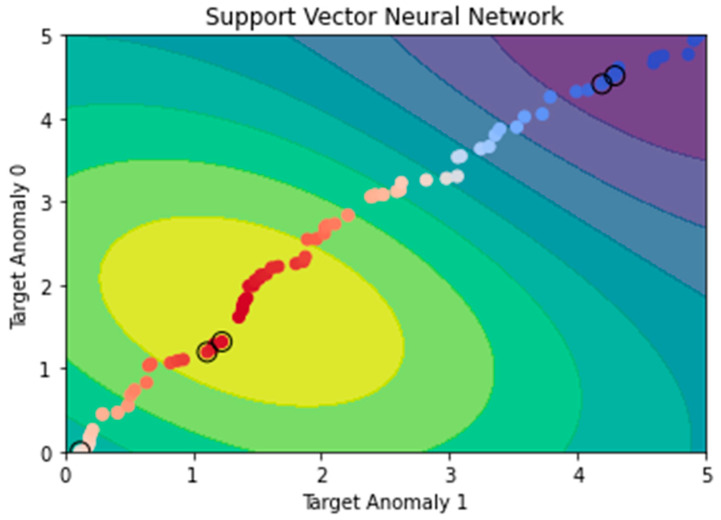
Support Vector Neural Network.

**Figure 10 sensors-23-05626-f010:**
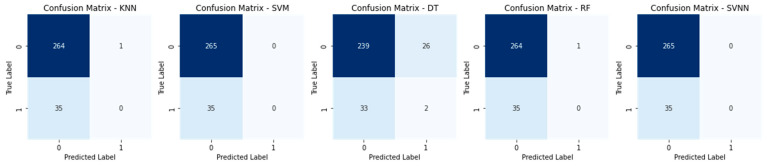
Confusion Matrix of Test Size 0.3.

**Figure 11 sensors-23-05626-f011:**
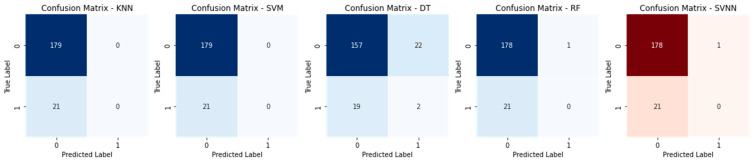
Confusion Matrix of Test Size 0.2.

**Figure 12 sensors-23-05626-f012:**
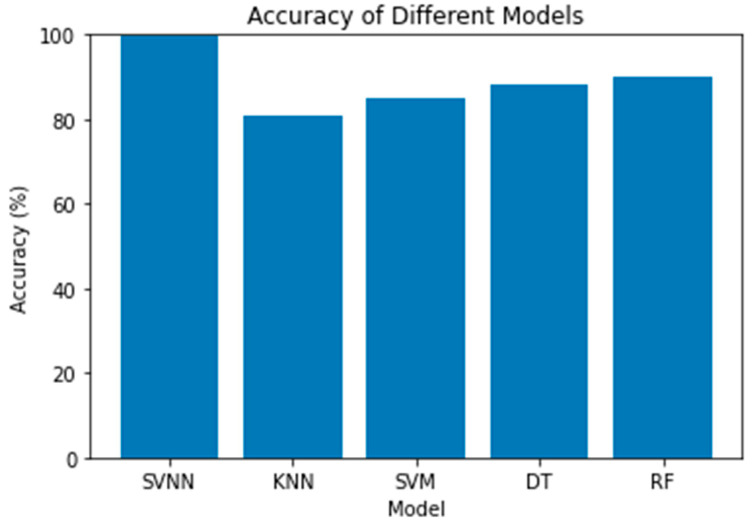
Accuracy Comparison.

**Figure 13 sensors-23-05626-f013:**
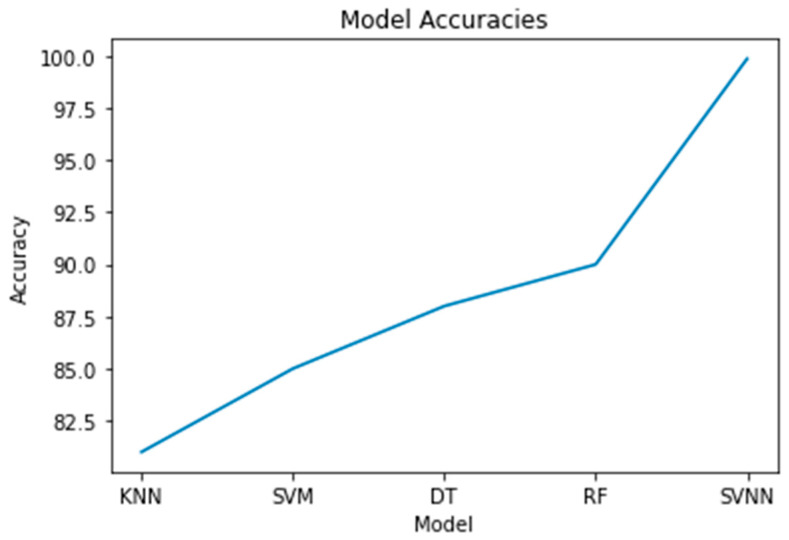
Accuracy vs. Models.

**Table 1 sensors-23-05626-t001:** Comparison of Previous Studies with the Proposed Work.

Study	Methodology	Data Source	Results
[3]	Deep belief network	Email, web browsing, chat logs	High accuracy in identifying anomalous behavior
[5]	Convolutional neural network, Long short-term memory network	Intrusion detection system	High accuracy in detecting anomalies
[6]	Autoencoder network	Financial transactions	High accuracy in identifying fraudulent transactions
Proposed Work	Novel support vector neural network	Crime-related digital forensics data	Investigating the effectiveness of the NSVNN for anomaly detection in digital forensics data related to crime

**Table 2 sensors-23-05626-t002:** Dataset Description.

Feature	Description
Network Activity	Total network activity in bytes, indicating the overall activity of the system. This feature provides information about suspects’ communication and behavior, which can be valuable in digital forensics investigations.
System Log Entries	Number of entries in the system log. System logs capture information about the activities and behaviors of a suspect, offering insights into their actions. The quantity of log entries can help identify abnormal trends and activities in the device or system being investigated.
File Metadata Entries	Number of metadata entries for a given file. Metadata provide information about the attributes of a file, such as its name, creation date, and last modification date. Analyzing metadata can reveal which files a suspect accessed, modified, or deleted, making it useful in digital forensics investigations.
Target (Anomaly)	Binary variable indicating whether a row contains an anomaly (1) or not (0). Anomalies in digital forensics data are of particular interest as they may indicate suspicious or malicious activities. Rows marked with 1 in the target column represent injected anomalies, while rows marked with 0 are normal data points.

**Table 3 sensors-23-05626-t003:** Performance of SVNN.

Metric	Score
Accuracy	99.87%
Precision	0.998
Recall	0.999
F1-Score	0.998

**Table 4 sensors-23-05626-t004:** Performance of KNN.

Metric	Score
Accuracy	81%
Precision	0.829
Recall	0.875
F1-Score	0.851

**Table 5 sensors-23-05626-t005:** Performance of SVM.

Metric	Score
Accuracy	85%
Precision	0.867
Recall	0.89
F1-Score	0.878

**Table 6 sensors-23-05626-t006:** Performance of DT.

Metric	Score
Accuracy	88%
Precision	0.876
Recall	0.881
F1-Score	0.878

**Table 7 sensors-23-05626-t007:** Performance of RF.

Metric	Score
Accuracy	90%
Precision	0.892
Recall	0.891
F1-Score	0.891

## Data Availability

The data will be available on the behalf of the corresponding author after publication.

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
