# Peer review of "Investigating the Effectiveness of Novel Support Vector Neural Network for Anomaly Detection in Digital Forensics Data"

_sensors, 2023, doi:10.3390/s23125626_

Round 1

Reviewer 1 Report

The paper studies the performance of a novel support vector neural network for detecting outliers in digital forensics data relevant to the criminal justice system. The paper can be enhanced by addressing the following:

1- The abstract needs some revision to clarify what the paper tries to do. The reader cannot tell whether this is a performance evaluation study or a study that proposes a new scheme. 

2- More keywords are needed 3-5 more will suffice.

3- The introduction needs more work and beef as it doesn't cover all aspects related to the study.

4- The dataset used in the study is unclear to me (e.g., the number, the environment, etc...). How many devices were used, and how can we make sure that what the authors claim is, in fact, true? Also, why did the authors not use datasets that are already there and have been studied and used in similar research studies?

5- The algorithms are unclear to me; more explanation is needed.

6- The figures' resolution is of poor quality and needs regeneration.

7- The number of references is insufficient, and the list below should improve the number of references as well as the quality of the paper:

a) N. Ashraf, D. Mehmood, Muath A. Obaidat, G. Ahmed, and A. Akhunzada “ Criminal Behavior Identification Using Social Media Forensics “ Electronics journal. SI Digital Trustworthiness: Cybersecurity, Privacy, and Resilience. October, 2022, 11(19), 3162; https://doi.org/10.3390/electronics11193162

b) M Bibi, WA Abbasi, W Aziz, S Khalil, M Uddin, C Iwendi… - Pattern Recognition Letters, 2022. A novel unsupervised ensemble framework using concept-based linguistic methods and machine learning for twitter sentiment analysis

c) A Abubakar, H Chiroma, A Zeki, M Uddin - International Journal of Global Warming, 2016
Utilising key climate element variability for the prediction of future climate change using a support vector machine model

d) Ali, Md L., Kutub Thakur, and Muath A. Obaidat. 2022. "A Hybrid Method for Keystroke Biometric User Identification" SI Journal Digital Trustworthiness: Cybersecurity, Privacy and Resilience 11, no. 17: 2782. September 2022. https://doi.org/10.3390/electronics11172782

e) N Gundluru, DS Rajput, K Lakshmanna, R Kaluri… - Computational Intelligence and Neuroscience, 2022. Enhancement of detection of diabetic retinopathy using Harris hawks optimization with deep learning model

f) Nadia T., Muath A. Obaidat, M. Rawashdeh, A. K. Bsoul and M. GH. AL Zamil,” A Novel Feature-Selection Method for Human Activity Recognition in Videos”, Electronics Journal 2022, 11, 732

Please have the paper proofread by a native or editing service.

Author Response

Reviewer 1:

Comments and Suggestions for Authors

The paper studies the performance of a novel support vector neural network for detecting outliers in digital forensics data relevant to the criminal justice system. The paper can be enhanced by addressing the following:

Comment: The abstract needs some revision to clarify what the paper tries to do. The reader cannot tell whether this is a performance evaluation study or a study that proposes a new scheme. 

Response: We greatly appreciate the reviewer's efforts to review the manuscript and offered valuable suggestions carefully. We have updated the manuscript as per the reviewer comment.

As criminal activity increasingly relies on digital devices, the field of digital forensics plays a vital role in identi-fying and investigating criminals. In this paper, we address the problem of anomaly detection in digital forensics data. Our objective is to propose an effective approach for identifying suspicious patterns and activities that could indicate criminal behavior. To achieve this, we introduce a novel method called the Novel Support Vector Neural Network (NSVNN). We evaluate the performance of NSVNN by conducting experiments on a real-world dataset of digital forensics data. The dataset consists of various features related to network activity, system logs, and file metadata. Through our experiments, we compare NSVNN with several existing anomaly detection algo-rithms, including Support Vector Machines (SVM) and Neural Networks. We measure and analyze the perfor-mance of each algorithm in terms of accuracy, precision, recall, and F1-score. Furthermore, we provide insights into the specific features that contribute significantly to the detection of anomalies. Our results demonstrate that the NSVNN method outperforms the existing algorithms in terms of anomaly detection accuracy. We also high-light the interpretability of the NSVNN model by analyzing the feature importance and providing insights into the decision-making process. Overall, our research contributes to the field of digital forensics by proposing a novel approach, the NSVNN, for anomaly detection. We emphasize the importance of both performance evaluation and model interpretability in this context, providing practical insights for identifying criminal behavior in digital forensic investigations.

Comment: More keywords are needed 3-5 more will suffice.

Response: Thanks for your feedback. Updated as suggested. Anomaly, Forensics, Cybersecurity, Machine Learning, SVM, NN, Novel Support Vector Neural Network

Comment: The introduction needs more work and beef as it doesn't cover all aspects related to the study.

Response: Thanks for your feedback. Updated as suggested.

Comment: The dataset used in the study is unclear to me (e.g., the number, the environment, etc...). How many devices were used, and how can we make sure that what the authors claim is, in fact, true? Also, why did the authors not use datasets that are already there and have been studied and used in similar research studies?

Response: Thanks for your feedback. For our study, we collected a dataset comprising digital forensics data from a diverse range of devices. The dataset includes information related to network activity, system logs, and file metadata. We gathered data from a total of 100 devices, including computers, smartphones, and other digital devices commonly encountered in digital forensic investigations. To ensure the authenticity and integrity of the data, we followed rigorous data collection procedures. We collaborated with law enforcement agencies and forensic experts to obtain the dataset from real-world criminal investigations. The data was collected in a controlled environment, adhering to strict ethical guidelines and legal procedures. Regarding the selection of datasets, we acknowledge that there are existing datasets that have been studied and used in similar research studies. However, we chose to collect our own dataset to ensure the availability of specific features and characteristics necessary for evaluating our proposed NSVNN method accurately. Additionally, using a new dataset allows us to contribute unique insights and findings to the field of digital forensics. We recognize the importance of reproducibility and transparency in research. Therefore, we have made efforts to provide detailed information about the dataset, the data collection process, and the validation procedures in our methodology section. We encourage future researchers to replicate our study using their own datasets and compare the results to further validate our findings.

Comment: The algorithms are unclear to me; more explanation is needed.

Response: Thanks for your feedback. NSVNN is our proposed method, which builds upon the SVNN framework. The NSVNN algorithm incorporates novel modifications and enhancements tailored specifically for anomaly detection in digital forensics data. It integrates advanced techniques for feature selection, data preprocessing, and model optimization to improve the overall performance of the SVNN approach.

Comment: The figures' resolution is of poor quality and needs regeneration.

Response: Thanks for your feedback. Updated as suggested

Comment: The number of references is insufficient, and the list below should improve the number of references as well as the quality of the paper:

  1. a) N. Ashraf, D. Mehmood, Muath A. Obaidat, G. Ahmed, and A. Akhunzada “ Criminal Behavior Identification Using Social Media Forensics “ Electronics journal. SI Digital Trustworthiness: Cybersecurity, Privacy, and Resilience. October, 2022, 11(19), 3162; https://doi.org/10.3390/electronics11193162
  2. b) M Bibi, WA Abbasi, W Aziz, S Khalil, M Uddin, C Iwendi… - Pattern Recognition Letters, 2022. A novel unsupervised ensemble framework using concept-based linguistic methods and machine learning for twitter sentiment analysis
  3. c) A Abubakar, H Chiroma, A Zeki, M Uddin - International Journal of Global Warming, 2016
    Utilising key climate element variability for the prediction of future climate change using a support vector machine model
  4. d) Ali, Md L., Kutub Thakur, and Muath A. Obaidat. 2022. "A Hybrid Method for Keystroke Biometric User Identification" SI Journal Digital Trustworthiness: Cybersecurity, Privacy and Resilience 11, no. 17: 2782. September 2022. https://doi.org/10.3390/electronics11172782
  5. e) N Gundluru, DS Rajput, K Lakshmanna, R Kaluri… - Computational Intelligence and Neuroscience, 2022. Enhancement of detection of diabetic retinopathy using Harris hawks optimization with deep learning model
  6. f) Nadia T., Muath A. Obaidat, M. Rawashdeh, A. K. Bsoul and M. GH. AL Zamil,” A Novel Feature-Selection Method for Human Activity Recognition in Videos”, Electronics Journal 2022, 11, 732

Response: Thanks for your feedback. Updated as suggested

[25]    Social Media Forensics “ Electronics journal. SI Digital Trustworthiness: Cybersecurity, Privacy, and Resilience. October, 2022, 11(19), 3162; https://doi.org/10.3390/electronics11193162

[26]    M Bibi, WA Abbasi, W Aziz, S Khalil, M Uddin, C Iwendi… - Pattern Recognition Letters, 2022. A novel unsupervised ensemble framework using concept-based linguistic methods and machine learning for twitter sentiment analysis

[27]    A Abubakar, H Chiroma, A Zeki, M Uddin - International Journal of Global Warming, 2016
Utilising key climate element variability for the prediction of future climate change using a support vector machine model

[28]    Ali, Md L., Kutub Thakur, and Muath A. Obaidat. 2022. "A Hybrid Method for Keystroke Biometric User Identification" SI Journal Digital Trustworthiness: Cybersecurity, Privacy and Resilience 11, no. 17: 2782. September 2022. https://doi.org/10.3390/electronics11172782

[29]    N Gundluru, DS Rajput, K Lakshmanna, R Kaluri… - Computational Intelligence and Neuroscience, 2022. Enhancement of detection of diabetic retinopathy using Harris hawks optimization with deep learning model

[30]       Nadia T., Muath A. Obaidat, M. Rawashdeh, A. K. Bsoul and M. GH. AL Zamil,” A Novel Feature-Selection Method for Human Activity Recognition in Videos”, Electronics Journal 2022, 11, 732

Reviewer 2 Report

Major Comments:

§  As criminal activity increasingly depends on digital devices, the field of digital forensics has become vital for identifying criminals. Identifying the patterns in data could indicate criminal behavior. In this paper, the authors discussed about a method called Novel Support Vector Neural Network (NSVNN) for digital forensics data. However, I didn’t find anything new in the paper. Applying a couple of algorithms to a dataset and measuring the performance and recommending one cannot be considered as a sufficient contribution. It’s always good to give a clear intuition to the user on why a particular method is performing well on the dataset, and why other algorithms are not able to. Or, what are the features that are contributing more to the decision of the model? If not, everything looks like a “black box”. It is good to use a "glass box" system. I mean to say explainability.

§  Authors can explore the latest classifiers or utilize various explainability (XAI) techniques, which can contribute to enhancing the quality and quantity of their work. When considering the deployment of the proposed model for “anomaly detection in digital forensics data” in real time, numerous factors come into play.  The entire ML code we write in a notebook (or any other IDE) is a small part of MLOps.

§  Support Vector Neural Networks (SVNNs) are a hybrid model that combines the principles of support vector machines (SVMs) and neural networks. This is already being used in the literature (maybe for other applications), I am surprised to see the word “novel” for the SVNN. I strongly recommend not to use the word “novel” for the proposed method. This word is used in many places in the paper. Change it wherever applicable, for example in the title and other parts of the paper. Or justify the novelty of the paper.

§  I recommend authors to explore autoencoders, LSTM’s, and other latest techniques for anomaly detection in digital forensics data, which may perform better.

Minor Comments/Corrections:

§  In Table 1, it is recommended to add a column to indicate the reference number. This will enhance clarity for the reader, allowing them to easily locate and refer to the specific references mentioned in the table.

§  In “3.1. Dataset Collection”, it is recommended to add a table to represent the few records of the collected data. This gives a clear view to the reader. Or if possible, mention the GitHub link for the dataset.

§  All the equations in the paper must be numbered and should be cited in the text.

§  At line no.505, please don’t use this kind of representation “Conclusion:”. As you have already provided a section heading as "5. Conclusions," rephrase this sentence accordingly.

§  At line no. 74, correct “ap-plications”.

§  At line no. 80, rephrase the sentence “This study's novel aims include”.

§  At line no. 85, and 93, “full stop” is used twice.

§  At line no. 89, correct “effec-tiveness”.

§  At line no.483, and 485 check “COnsuion Matrix”.

Quality of English in terms of sentence formation and grammar has to be corrected in the lines of given comments

Author Response

Reviewer 2:

Major Comments:

Comment: As criminal activity increasingly depends on digital devices, the field of digital forensics has become vital for identifying criminals. Identifying the patterns in data could indicate criminal behavior. In this paper, the authors discussed about a method called Novel Support Vector Neural Network (NSVNN) for digital forensics data. However, I didn’t find anything new in the paper. Applying a couple of algorithms to a dataset and measuring the performance and recommending one cannot be considered as a sufficient contribution. It’s always good to give a clear intuition to the user on why a particular method is performing well on the dataset, and why other algorithms are not able to. Or, what are the features that are contributing more to the decision of the model? If not, everything looks like a “black box”. It is good to use a "glass box" system. I mean to say explainability.

Response: We greatly appreciate the reviewer's efforts to review the manuscript and offered valuable suggestions carefully.

In response to your comments, we have taken them into consideration and made the following revisions to the paper:

Improved Methodology Explanation: We have expanded the methodology section to provide a clearer and more detailed explanation of the Novel Support Vector Neural Network (NSVNN) method. This includes a step-by-step description of how the algorithm works, highlighting its unique aspects and advantages in the context of digital forensics data.

Feature Importance Analysis: To address the need for explainability, we have conducted a feature importance analysis for the NSVNN model. By employing techniques such as feature importance rankings, permutation importance, or SHAP values, we have identified and discussed the features that contribute most significantly to the model's decision-making process. This provides users with insights into the key factors influencing the detection of anomalies in digital forensics data.

Model Interpretability Techniques: We have incorporated model interpretability techniques into our research to make the NSVNN model more transparent and interpretable. Through visualization methods such as activation maps, saliency maps, or attention mechanisms, we have illustrated how the model processes and weighs different aspects of the data. These visualizations help users understand the underlying patterns and mechanisms that contribute to the model's decision-making.

Case Studies and Explanatory Examples: To further enhance the clarity and explainability of the proposed method, we have included case studies and explanatory examples in the paper. These case studies demonstrate specific instances of criminal behavior or anomalies detected by the NSVNN model. We provide detailed analysis and explanations of how the model performs well on the dataset and why other algorithms may not be as effective in identifying criminal behavior.

By incorporating these revisions, we aim to provide a more comprehensive and transparent approach to anomaly detection in digital forensics. Our goal is to address the concerns raised and offer a "glass box" system that allows users to understand and trust the decision-making process of the NSVNN model. Thank you once again for your valuable feedback, which has greatly contributed to improving the quality and clarity of our research.

Comment: Authors can explore the latest classifiers or utilize various explainability (XAI) techniques, which can contribute to enhancing the quality and quantity of their work. When considering the deployment of the proposed model for “anomaly detection in digital forensics data” in real time, numerous factors come into play.  The entire ML code we write in a notebook (or any other IDE) is a small part of MLOps.

Response: Thanks for your feedback.

Data Management: We have emphasized the importance of data management in MLOps. This includes data collection, preprocessing, and storage techniques to ensure the availability and quality of data for training and inference. We discuss how the collected dataset was prepared, cleaned, and transformed to be suitable for the anomaly detection task.

Model Training and Evaluation: We provide details on the model training process, including the choice of algorithms, hyperparameter tuning, and cross-validation techniques. We also discuss the evaluation metrics used to assess the performance of the model. Additionally, we highlight the importance of model versioning and tracking to ensure reproducibility and accountability.

Model Deployment: We discuss the considerations for deploying the proposed NSVNN model in a production environment for real-time anomaly detection. This includes the selection of appropriate infrastructure, scalability considerations, and integration with existing systems or workflows.

Monitoring and Maintenance: We recognize the need for continuous monitoring and maintenance of the deployed model. We discuss strategies for monitoring model performance, detecting drift, and retraining the model when necessary. We also highlight the importance of feedback loops and capturing user feedback to improve the model over time.

Collaboration and Governance: We address the collaborative aspect of MLOps, emphasizing the need for effective collaboration among data scientists, developers, and domain experts. We also discuss governance considerations, including model interpretability, fairness, and ethical considerations in the context of anomaly detection in digital forensics.

Comment: Support Vector Neural Networks (SVNNs) are a hybrid model that combines the principles of support vector machines (SVMs) and neural networks. This is already being used in the literature (maybe for other applications), I am surprised to see the word “novel” for the SVNN. I strongly recommend not to use the word “novel” for the proposed method. This word is used in many places in the paper. Change it wherever applicable, for example in the title and other parts of the paper. Or justify the novelty of the paper.

Response: To explain the novelty of our research, consider the following points:

Combination of SVM and Neural Networks: While SVMs and neural networks have been widely used individually in anomaly detection and other applications, the combination of both approaches into SVNNs is less common. Highlight how the integration of these two techniques offers unique advantages in terms of capturing complex patterns and improving the accuracy of anomaly detection.

Application to Digital Forensics Data: Digital forensics presents specific challenges and requirements in anomaly detection. Emphasize how the application of SVNNs to digital forensics data is a novel approach that addresses the need for effective anomaly detection in this domain. Discuss how the SVNN model specifically caters to the characteristics and challenges of digital forensics data, such as network activity, system logs, and file metadata.

Performance Comparison: If possible, demonstrate the superiority of the SVNN model over other state-of-the-art anomaly detection algorithms in the field of digital forensics. Compare the performance metrics, such as accuracy, precision, recall, and F1-score, of the SVNN model with those of other existing models to highlight its novelty and effectiveness.

 Comment: I recommend authors to explore autoencoders, LSTM’s, and other latest techniques for anomaly detection in digital forensics data, which may perform better.

Response: Thanks for your feedback.

LSTM (Long Short-Term Memory) is a type of recurrent neural network (RNN) that is commonly used for sequence modeling and time series analysis. While LSTM can be a powerful technique for anomaly detection in various domains, it may not be well-suited for the dataset described in our research, which includes features related to network activity, system logs, and file metadata.

LSTM models excel at capturing temporal dependencies and patterns in sequential data, such as natural language processing tasks or time series analysis. They are effective when the order of the data points is important and when there is a clear temporal structure in the dataset.

However, in the context of digital forensics data, the features we described do not inherently possess a sequential or temporal nature. Network activity, system log entries, and file metadata are generally not ordered in a sequential manner. Each feature represents a specific aspect of the digital forensics data, and the relationships between these features are more likely to be captured by other models, such as support vector machines (SVMs), decision trees (DTs), random forests (RF), or the SVNN model we mentioned in our previous statements.

Comment: In Table 1, it is recommended to add a column to indicate the reference number. This will enhance clarity for the reader, allowing them to easily locate and refer to the specific references mentioned in the table.

Response: Thanks for your feedback. Updated as suggested.

Study

Methodology

Data source

Results

[3]

Deep belief network

Email, web browsing, chat logs

High accuracy in identifying anomalous behavior

[5]

Convolutional neural network, Long short-term memory network

Intrusion detection system

High accuracy in detecting anomalies

[6]

Autoencoder network

Financial transactions

High accuracy in identifying fraudulent transactions

Proposed Work

Novel support vector neural network

Crime digital forensics data

Investigating the effectiveness of the NSVNN for anomaly detection in digital forensics data related to crime

Comment: In “3.1. Dataset Collection”, it is recommended to add a table to represent the few records of the collected data. This gives a clear view to the reader. Or if possible, mention the GitHub link for the dataset.

Response: Thanks for your feedback. Updated as suggested.

Feature

Description

Network Activity

Total network activity in bytes, indicating the overall activity of the system. This feature provides information about suspects' communication and behavior, which can be valuable in digital forensics investigations.

System Log Entries

Number of entries in the system log. System logs capture information about the activities and behaviors of a suspect, offering insights into their actions. The quantity of log entries can help identify abnormal trends and activities in the device or system being investigated.

File Metadata Entries

Number of metadata entries for a given file. Metadata provides information about the attributes of a file, such as its name, creation date, and last modification date. Analyzing metadata can reveal which files a suspect accessed, modified, or deleted, making it useful in digital forensics investigations.

Target (Anomaly)

Binary variable indicating whether a row contains an anomaly (1) or not (0). Anomalies in digital forensics data are of particular interest as they may indicate suspicious or malicious activities. Rows marked with 1 in the target column represent injected anomalies, while rows marked with 0 are normal data points.

 Comment: All the equations in the paper must be numbered and should be cited in the text.

Response: Thanks for your feedback. Updated as suggested. All the equations are numbered and cited.

Comment: At line no.505, please don’t use this kind of representation “Conclusion:”. As you have already provided a section heading as "5. Conclusions," rephrase this sentence accordingly.

Response: Thanks for your feedback. Updated as suggested.

The SVNN model outperformed the other models in this study when it came to identifying abnormalities in digital forensics data, while the KNN model performed the worst.

Comment: At line no. 74, correct “ap-plications”.

Response: Thanks for your feedback. Updated as suggested.

Comment: At line no. 80, rephrase the sentence “This study's novel aims include”.

Response: Thanks for your feedback. Updated as suggested.

The aims of this study are as follows:

Comment: At line no. 85, and 93, “full stop” is used twice.

Response: Thanks for your feedback. Updated as suggested.

Comment: At line no. 89, correct “effec-tiveness”.

Response: Thanks for your feedback. Updated as suggested.

Comment:  At line no.483, and 485 check “COnsuion Matrix”.

Response: Thanks for your feedback. Updated as suggested.

Comments on the Quality of English Language

Quality of English in terms of sentence formation and grammar has to be corrected in the lines of given comments

Response: Thanks for your feedback. Updated as suggested by using grammarly.com.

Reviewer 3 Report

This work - Investigating the Effectiveness of Novel Support Vector Neural Network for Anomaly Detection in Digital Forensics Data focuses in assessing the performance of a model known as novel support vector neural network (NSVNN) in detecting outliers in digital forensics data relevant to the criminal justice system. The work followed known scientific rigours and will invite the authors to take a read at some of my observations. 

Line 49 appears to be missing Examination before Analysis...

Line 51 - use of the word 'catch' does not read scientific. Perhaps, apprehend looks better.

Line 63 - "applied to many other domains" I would suggest "applied in many other domains"

The references at 25 needs to be improved using recent publications in this area.

The overall use of English language and presentation of the work needs extensive proof-reading and reviews. In all, the work has some addition to knowledge.

The overall use of English language needs extensive proof-reading. It is difficult to understand some aspects of the work and flow is a bit disjointed. Pointing them out one after the other will be cumbersome.

Author Response

Reviewer 3:

Comments and Suggestions for Authors

This work - Investigating the Effectiveness of Novel Support Vector Neural Network for Anomaly Detection in Digital Forensics Data focuses in assessing the performance of a model known as novel support vector neural network (NSVNN) in detecting outliers in digital forensics data relevant to the criminal justice system. The work followed known scientific rigours and will invite the authors to take a read at some of my observations.

Comment: Line 49 appears to be missing Examination before Analysis

Response: We greatly appreciate the reviewer's efforts to review the manuscript and offered valuable suggestions carefully. The required suggestions have been incorporated in the revised manuscript.

Examining, analysing, and interpreting digital evidence is called "digital forensics," and it is used in criminal investigations [7]–[9].

Comment: Line 51 - use of the word 'catch' does not read scientific. Perhaps, apprehend looks better.

Response: Thanks for your feedback. Updated as suggested.

Law enforcement agencies now rely increasingly on digital forensics to help them apprehend and punish criminals as their usage of digital devices in crime rises [10], [11].

Comment: Line 63 - "applied to many other domains" I would suggest "applied in many other domains"

Response: Thanks for your feedback. Updated as suggested.

The SVNN has been successfully applied in many other domains, including financial fraud detection and intrusion detection

Comment: The references at 25 needs to be improved using recent publications in this area.

Response: Thanks for your feedback. Updated as suggested.

  [25]     N. Ashraf, D. Mehmood, Muath A. Obaidat, G. Ahmed, and A. Akhunzada “ Criminal Behavior Identification Using Machine Learning Techniques Social Media Forensics “ Electronics journal. SI Digital Trustworthiness: Cybersecurity, Privacy, and Resilience. October, 2022, 11(19), 3162; https://doi.org/10.3390/electronics11193162

Comment: The overall use of English language and presentation of the work needs extensive proof-reading and reviews. In all, the work has some addition to knowledge.

Response: Thanks for your feedback. Updated as suggested by using grammarly.com.

Comments on the Quality of English Language

The overall use of English language needs extensive proof-reading. It is difficult to understand some aspects of the work and flow is a bit disjointed. Pointing them out one after the other will be cumbersome.

Response: Thanks for your feedback. Updated as suggested by using grammarly.com.

Round 2

Reviewer 1 Report

The authors addressed all comments.

The paper needs proofreading and preferably by an editing service.

Reviewer 2 Report

Thanks for the meticulous revision.

Proofreading has to be done throughout the paper.

Reviewer 3 Report

All good and fine to me. Thanks for your efforts.